# In Silico Modelling of Neurodegeneration Using Deep Convolutional Neural Networks

## Abstract

Although current research aims to use and improve deep learning networks by applying knowledge about the structure and function of the healthy human brain and vice versa, the potential of using such networks to model neurodegenerative diseases remains largely understudied. In this work, we present a novel feasibility study modeling dementia in silico with deep convolutional neural networks. Therefore, deep convolutional neural networks were fully trained to perform visual object recognition, and then progressively injured in two distinct ways. More precisely, damage was progressively inflicted mimicking neuronal as well as synaptic injury. Synaptic injury was applied by randomly deleting weights in the network, while neuronal injury was simulated by removing full nodes or filters in the network. After each iteration of injury, network object recognition accuracy was evaluated. Saliency maps were generated using the uninjured and injured networks and quantitatively compared using the structural similarity index measure for test set images to further investigate the loss of visual cognition. The quantitative evaluation revealed cognitive function of the network progressively decreased with increasing injury load. This effect was more pronounced for synaptic damage. As damage increased, the model focus shifted away from the main objects in the images and became more dispersed. This shift in attention was quantitatively evidenced by a decrease in the structural similarity index measure comparing the saliency maps of corresponding uninjured and injured models, as a function of injury. The results of this study provide a promising foundation to develop in silico models of neurodegenerative diseases using deep learning networks. The effects of neurodegeneration found for the in silico model are especially similar to the loss of visual cognition seen in patients with posterior cortical atrophy.

## 1 Introduction

Amidst the current explosion of big data, deep learning models have emerged as integral tools for solving many complex classification, regression, and object recognition problems [Lo Vercio et al., 2020]. More recently, deep convolutional neural networks (DCNNs) are also increasingly explored as potential tools to model information processing in the mammalian brain [Yamins and DiCarlo, 2016]. This is assumed possible because DCNNs were originally inspired by the neuron and synaptic structure found in the mammalian visual cortex [Rawat and Wang, 2017]. To date, studies have explored similarities in neural activations between DCNNs and primate brains, and have reported positive correlations between responses in specific areas of these models and the primate ventral visual stream[Yamins et al., 2014].

Currently, machine learning research primarily aims to advance the biological similarity of DCNNs to produce more brain-like artificial neural network models with the hope of improving their task-

Submitted to 3rd Workshop on Shared Visual Representations in Human and Machine Intelligence (SVRHM 2021) of the Neural Information Processing Systems (NeurIPS) conference.

specific performance. Meanwhile, computational neuroscience research is primarily interested in using DCNNs as a computational model of the healthy brain [Yamins and DiCarlo, 2016, Kriegeskorte, 2015, Richards et al., 2019, Peters and Kriegeskorte, 2021]. In this research project, we explored the potential use of employing DCNNs as in silico models of neurodegenerative diseases, a largely unexplored research direction. Specifically, this work provides one of the first proof of concepts of an in silico model of posterior cortical atrophy (PCA). PCA is a disorder associated with Alzheimer's disease and is characterized by visual dysfunction such as visual agnosia and simultanagnosia [da Silva et al., 2017]. In the case of visual agnosia, patients lose the ability to visually recognize and identify familiar objects without losing the ability to see the object. Simultanagnosia is marked by failure to perceive multiple visual locations simultaneously or to shift attention from one object to another. PCA is caused by the accelerated degeneration and thinning of the associated visual cortices (i.e., V1, V2, V3, V4). Since DCNNs were specifically designed for object recognition and modelled following information processing in the mammalian brain, neuronal injuries as seen in PCA can be intuitively modelled in DCNNs. In this work, synaptic damage was performed by randomly removing weights in the trained network, while neuronal damage was modelled by randomly removing nodes, including all connecting weights. The effect of the two injury types on visual object recognition capabilities was quantitatively and qualitatively analyzed by assessing model accuracy and structural differences in saliency maps between healthy and injured models.

## 2 Methods and materials

### 2.1 Models and data

This work is based on the VGG19 model pretrained on the ImageNet database described in more detail by Russakovsky et al., 2015, which was fine-tuned on the Imagenette database [Russakovsky et al., 2015, Howard]. The VGG19 model was selected for this purpose as it has one of the highest correlation values when compared to mammalian neuronal activation data, measured using the Brain-Score [Schrimpf et al., 2018]. This model contains 16 convolutional layers, with each convolutional block followed by a max-pooling layer. The final four layers are fully-connected dense layers; the first two containing 1024 neurons, the third 512 neurons, finally followed by a 10-dimensional softmax classification layer. The network was optimized using the Adam optimizer and a learning rate of 0.001. No drop-out was used in the fine tuning of the additional three dense layers. We separately trained 25 VGG19 models, each initialized with a different set of weights in the dense layers to reduce potential biases.

Imagenette is a smaller subset of the full ImageNet database and consists of ten easily identifiable classes containing both animate and inanimate objects. The train-test split used in this work consisted of 9469 and 3925 images, respectively. Images were scaled to dimensions of 224×224×3. Prior to damaging the network, the fine-tuned VGG19 performed object classification on the Imagenette test set with an accuracy of 94.2% $\pm$ 0.006% when averaged across all 25 models. All 25 initial models were subjected to increasing rates of progressive synaptic or neuronal injury.

### 2.2 Neurodegeneration - Simulated post cortical atrophy

Synaptic damage was inflicted on the baseline trained models by randomly setting x percent of the weights in the model to zero, effectively severing connections between neurons in the model, which simulates synaptic injury. The selection of weights that were injured was randomly generated 25 times, one for each of the 25 models, to reduce the potential bias introduced by the randomization process. In each iteration, 1% additional damage was increasingly applied to simulate progressive damage. In a second set of experiments, neuronal injury was modelled by progressively removing entire nodes from convolutional layers and dense layers of the network. In the convolutional layers, nodes are equivalent to filters and in dense layers, a node was considered a unit. When a node was removed, all adjacent weights were effectively deleted. Neuronal injury was randomly dispersed throughout all convolutional and dense layers and progressively increased with 1% increments.

### 2.3 Saliency maps

In order to further investigate the cognitive decline in the injured networks, saliency maps were generated for each iteration of generated injury and compared for all test set images between

uninjured and injured networks. Saliency maps are frequently used in computer vision tasks to enhance understanding around which parts of an input stimuli a DCNN focuses on to arrive at a classification [Simonyan et al., 2014]. In this research, GradCam saliency maps were generated for every image in the test set at each injury level. GradCam computes the gradients of the class score with respect to activations of the last convolutional block of the network. In this work, the experiments used the predicted class as the class score. The structural similarity index measure (SSIM) was calculated between the healthy network saliency maps and those generated by the injured networks as a means to quantify the shift in attention the network exhibits as a function of injury. SSIM is commonly used and a well-accepted metric to compare similarity between images [Bylinskii et al., 2019, Wang et al., 2004].

# 3 Results

## 3.1 Object recognition accuracy

Synaptic injury nearly immediately led to a decrease in model accuracy. The steepest decline in object recognition accuracy was seen between 13% and 23% synaptic injury, while at injury levels of 30% and greater, the model performed at chance level of 10% (see Figure 1A). In contrast to this finding, model performance retained an object recognition accuracy greater than 90% with neuronal injury until it reached damage levels of 79%. The steepest loss of accuracy for neuronal injury occurred between 87% and 99% injury. It should be noted that even at 99% injury, the model performed considerably better than chance level (see Figure 1B).

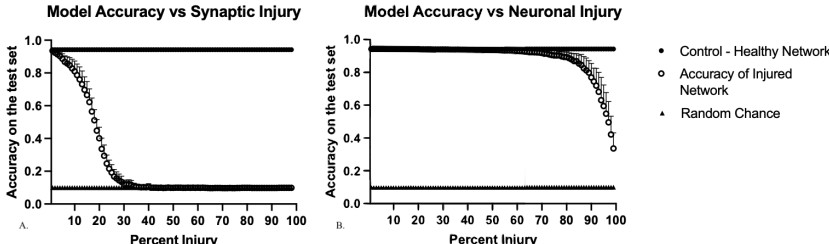

Figure 1: A) Model object recognition accuracy as the model underwent 1% increments of progressive synaptic injury. After 30% injury, the model performed at chance level. B) Model accuracy as progressive neuronal injury was applied. Model performance does not begin to degrade significantly until 65% damage. Data are presented as the mean + SD across all runs and all images in the test set.

## 3.2 Saliency maps using predicted class labels

Visual analysis of saliency maps revealed that attention of the uninjured model was correctly focused on sections of the test images that contribute meaningfully to the correct classification (see Figure 2). As synaptic damage increased, the focus of the model subtly began to shift away from the relevant objects in the images (see Figure 2A). These qualitative results are supported by the quantitative results that revealed decreasing structural similarity index measures (SSIM) with increasing injury (see Figure 2B) comparing the saliency maps of the uninjured networks to the corresponding saliency maps of the injured networks. When calculated and averaged across all images in the test set, the average SSIM was reduced from 1.0 to $0.348 \pm 0.016$ after the first 10% of synaptic injury. Once the model was unable to correctly classify which type of object is in a given input stimulus, the ability to focus on the relevant parts of the image was largely hindered. The activations within the network no longer maximized the probabilities of the correct classes. This impaired result was qualitatively evident in the 50% injured saliency map shown in Figure 2. It is also represented in Figure 2C where average SSIM is displayed as a function of model accuracy.

This increasing dissimilarity was much less qualitatively evident when progressive neuronal damage was applied (see Figure 3A). Upon visual inspection, the model appeared to retain some accuracy in attention focus on the given input stimuli, even at 90% neuronal injury. This retention of attention accuracy was also reflected in the average SSIM at 90% injury ($0.416 \pm 0.027$) (Figure 3B). While

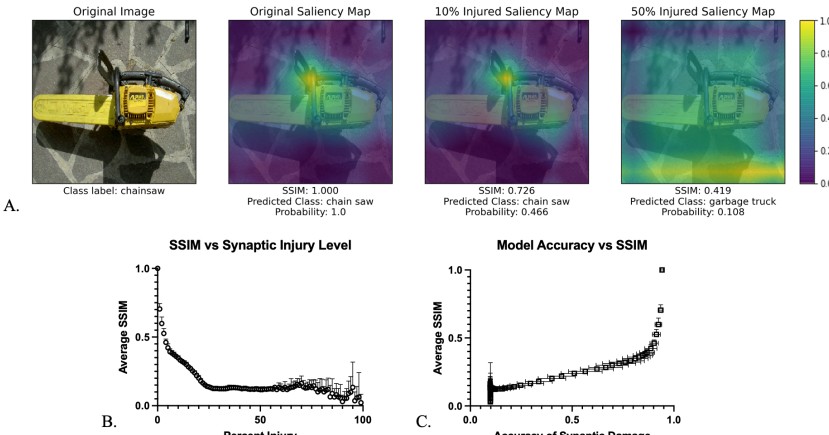

Figure 2: Attention focus of the model quantified using saliency maps are generated with respect to the predicted class label. Data are presented as mean + SD across all runs for all images in the test set. A) Qualitative examination of saliency maps at separate levels of synaptic injury. B) SSIM calculated between saliency maps as a function of synaptic injury. Between 1% and 20% injury, the SSIM is severely affected (Identical images are computed at SSIM=1). C) SSIM plotted as a function of model accuracy.

these effects of degeneration in saliency map similarity and thus, attention focus, were much less pronounced in neuronal injury, they are consistent with respect to overall model accuracy, as seen in the similarity between Figure 2C and Figure 3C.

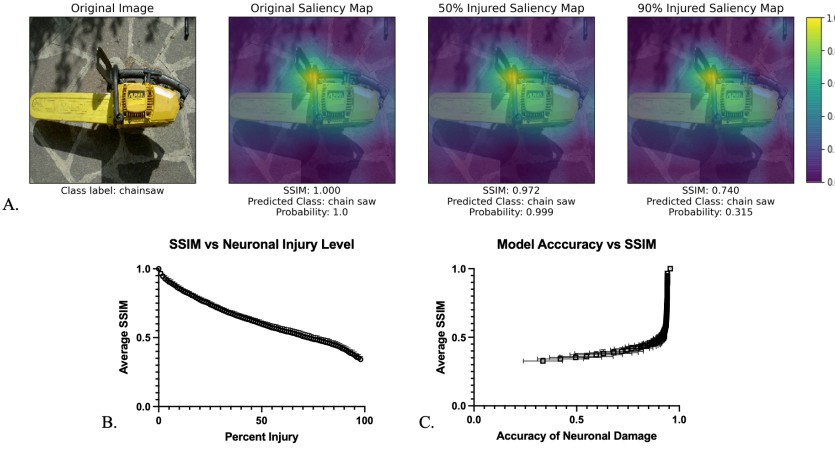

Figure 3: Attention focus of the model quantified using saliency maps are generated with respect to the predicted class as the neuronal injury is progressively applied. Data are presented as mean + SD. A) Qualitative examination of saliency maps at separate levels of neuronal injury. B) Changes in SSIM between saliency maps as injury is applied. SSIM gradually decreases. C) Average SSIM as a function of accuracy as neuronal damage is progressively applied to the network.

## 4   Discussion

### 4.1   Main Findings

The main finding of this study is that all models eventually become more cognitively impaired with respect to their object recognition abilities with progressively increasing amounts of injury. This relationship is analogous to cognitive decline seen in patients affected by neurodegenerative diseases,

such as Alzheimer's disease, who experience a loss of object recognition capabilities [Fox et al., 1999, Hodges et al., 1995, Jefferson et al., 2006]. Within this context, previous studies have shown that patients with Alzheimer's disease perform poorly on visual search tasks due to inefficiency in shifting attention to relevant targets as well as inefficiency in processing information held within a target [Tales et al., 2004]. Indeed, our preliminary research in modelling the onset of the neurodegeneration of PCA using deep learning models show that DCNNs behave similarly to biological neural networks in this respect.

The difference in how injury was imposed, i.e., synaptically or neuronally, provides crucial insight into the development of in silico models of biological phenomena using deep learning models. The fragility of the network when exposed to weight-based (synaptic) injury was highlighted in the severe decline of model accuracy and attention focus, even at rather small injury levels. When removing weights in a randomly dispersed manner in these static, feed-forward networks, the filters in the subsequent layers received a widespread lack of meaningful information. Thus, the poor information quickly affected the nodes in subsequent layers and, hence, the network's recognition capabilities. This is consistent with biological findings, in that synaptic loss results in less coordinated brain activity and may be the ultimate correlate to cognitive deficits due to Alzheimer's disease [John and Reddy, 2021, Kashyap et al., 2019].

Contrary to the effects of simulated synaptic injury, our results suggest that neuronal pruning has less severe effects on the qualitative and quantitative metrics investigated. A likely explanation for this finding is that removing a filter from a convolutional layer or a complete unit (neuron) from a layer in a deep learning network does not leave the subsequent layers with as much of a lack of information. Large convolutional neural networks, such as the VGG19, have proven to be quite robust in model compression studies, implying a certain level of redundancy in the network [Han et al., 2016]. The neuronal damage results we obtained in this study are similar to what previous machine learning literature on network pruning has reported [Hu et al., 2016]. In this stream of experiments, we observed that object recognition capabilities mostly remain at a high level until damage levels of 65% and greater are imposed. These results combined with the slow but yet progressive loss of object recognition accuracy is analogous to what patients suffering from PCA experience as a progressive loss of visuospatial and visuoperceptual skills [Crutch et al., 2012]. As Alzheimer's disease is most often the underlying pathology of PCA, it has been shown that clinical symptoms of Alzheimer's disease, such as visual cognitive decline, only present when substantial atrophy has occured [Fox et al., 1999]. The initial robustness of the human brain to injury is largely due to the extensive number of redundant connections that are in place to protect the system from structural breakdown [Kashyap et al., 2019]. This type of relationship is consistent and directly evident in the results of the in silico neuronal damage modelled in this study.

## 4.2 Limitations and future research directions

The main limitations of this study include the limited dataset that only contains ten classes of relatively easily categorizable objects. In order to more accurately model human visual cognition, a larger dataset with a more diverse range of class objects will be investigated in the future. Another limitation is the explicit difference between the strictly feed-forward structure of the DCNNs used in this work and the complex information processing that occurs in biological neural networks. Furthermore, the complexities of individual tau patterns and neurodegeneration resulting in different clinical symptoms, such as cognitive decline, are still not fully understood [Han et al., 2016]. Thus, building a generalized model of neurodegeneration and the subsequent cognitive deficits faces similar challenges. The development of this field of work has the potential to lead to positive societal implications by increasing understanding around the progression of these diseases. There are currently no foreseen negative impacts.

Crucial future research directions will be to incorporate model retraining between each iteration of injury to more accurately capture the inherent neuroplasticity in the degenerating human brain. It is expected that this will alleviate some of the extreme results seen in the synaptic injury simulation as weights will be able to update and compensate for certain initial amounts of damage. Additionally, more detailed evaluation metrics can be employed. Examining optimized networks with little to no redundancy will allow for further investigation into the effects of removing individual nodes or weights in the network. Finally, a future experimentation will include investigating focal and more concentrated neuronal loss, rather than randomly dispersed injury as applied in this study.

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
