# OpenReview forum: "In Silico Modelling of Neurodegeneration Using Deep Convolutional Neural Networks"
_NeurIPS.cc/2021/Workshop/SVRHM — SVRHM 2021 Poster_

### Official Review · Reviewer_zxin · 2021-10-19
**Review of "In Silico Modelling of Neurodegeneration Using Deep Convolutional Neural Networks"**

**Rating:** 4
**Confidence:** 3

**Review:**

## Summary:

In the present paper the authors provide first steps towards modelling neurodegenerational diseases in the human brain by using deep neural networks. They use the deep convolutional neural network VGG-19 as their target model finetuned on the ImageNette dataset. In order to model the effects of neurodegeneration in VGG-19 they randomly delete either a certain percentage of weights (so called "synaptic damage") or a certain percentage of nodes (so-called "neuronal damage"). The authors show that accuracy of the network on an object-categorization task decreased when progressively increasing the percentage of synaptic or neuronal damage in the network. Interestingly, the results showed a difference between the effects of synaptic damage and neuronal damage. Neuronal damage did only effect task performance at higher percentages of removal while synaptic damage affected task performance already at lower percentages. The authors also show that neuronal and synaptic damage was accompanied with changes in saliency maps which they link to attentional deficits found in humans suffering from neurodegenerational diseases.

## Pros:

- the paper addresses an important question an tries to pave the way for future studies using deep neural networks as models of brain dysfunction
- the paper is clearly written and the main results are concisely communicated
- the difference between synaptic and neuronal damage seems to be an surprising finding and invites further investigation

## Cons:

- I would be interested to read more about the differences between synaptic damage and neuronal damage, for example what is the effective difference in terms of number of removed parameters in the model for the two types of removal ? Can differences in performance be explained by differences in number of removed parameters ?
- I was wondering why the authors chose to not use dropout for finetuning the network. Could it be the case that this biases your networks to be more prone to decreases in task performance because of removal of single weights ? Can differences between neuronal and synaptic damage be explained by this choice ?
- the observation that saliency maps and SSIM metrics are different in a damaged network compared to the full network could be simply related to task performance and from the results it is not clearly distinguishable if the differences in these metrics are a result of decreased performance or if they explain the decreases in performance. However, I admit that I am not very familiar with the saliency map approach and the SSIM metric, and therefore I am less confident about this evaluation.
- On a more conceptual level, in my opinion the question the paper addresses is not specific enough to reveal something meaningful about brain dysfunction. As a far as I understand the paper answers the question if random damage to a model of the brain leads to decreases in performance. It might be argued that removing parts of any model will result in decreases in performance, regardless if the model is a good model of the brain and if the damage that was applied to the model is in any way related to damage that is found in the brain. While deep neural networks might indeed be useful for modelling neurodegenerational diseases, the demonstration that they also show decreased performance when they are damaged does not show that they are appropriate to model neurogenerational disease. To do this, theories about how neurodegenerational diseases affect brain function and behavior could be formulated and implemented in neural networks models and alternative hypotheses could be tested. The present paper, however, does not accomplish to test more specific hypotheses than the one proposing that any damage in both the brain and neural network models leads to reduced performance and changes in metrics of feature importance. Therefore, I consider the insight gained from the results of the paper as limited.

## Overall evaluation:

Overall, I cannot fully recommend to accept the presented paper for publication. Even though the analyses in the paper are presented clearly, the results follow a clear logic and are understandable, the authors do not address more specific questions about the mechanisms of neurodegenerational diseases beyond the relation between damage to a system and decreases in performance and changes in feature importance. In my opinion, this contribution is too marginal for acceptance at the workshop. However, I think that the paper could be improved if more details about the differences between neuronal and synaptic damage would be included (e.g. effective decrease in parameters, effect of dropout).

---

### Official Review · Reviewer_Gmo8 · 2021-10-29
**Interesting new avenue to understand neurodegenerative diseases through DNN modeling**

**Rating:** 7
**Confidence:** 4

**Review:**

Inspired by the use of deep neural networks as computational models of visual processing, the authors propose to not only treat these models as models for intact visual processing but also to explore their benefits for studying visual impairments resulting from neurological conditions such as posterior cortical atrophy (PCA). I appreciate this interesting approach and am curious to see where it will go. Overall, the paper is clearly written, appears methodologically sound and is an interesting start in this direction.

Comments:
- The authors draw parallels between their models decline in performance and neurodegenerative decline. They state that their main finding is that "all models eventually become more cognitively impaired with respect to their object recognition abilities with progressively increasing amounts of injury". I am wondering in how far this pattern of results may be specific to these model manipulations? For instance, I could gradually add dropout or noise to the network layers during inference and will likely also observe a similar decay in performance. In my view, to make a convincing link between the neurodegenerative decline and the specific network manipulation, it is necessary to show that this is specific to this manipulation or disease (incl. that this does not occur with other types of network perturbations or neurological conditions). One avenue to achieve this in future versions of this work could be to identify a more specific behavioural correlate or neural pattern to model, rather than just a general decline in object recognition performance that is also easily matched by other manipulations. I realize that this comment requires drastic changes to this project and it is not a recommendation for this workshop paper, but rather intended for future versions of this project.
- Does the x-axis in figure 1 really imply that the entire network has been affected at 100%? In the case of synaptic injury, how can the model then still perform highly above chance if all of its nodes have been taken out? Did the authors also take out the bias term when taking out a node?
- A related question, if 100% of all synaptic connections are affected is that then not equivalent to affecting 100% of the units? The data in Figure 1 and authors suggest a marked difference. It would be helpful to clarify this aspect in the manuscript.
- A minor suggestion on the language: Use intact rather than uninjured?
- Finally, while I like the idea of inspecting the model's saliency maps to understand how processing may break down, I'd caution the authors to use the term attention for both DNNs (in the context of the saliency maps) and patients (failure to deploy of 'top-down' attention) without additional reflection on the difference in the term as used in Psychology and ML (for a discussion on this see, for instance, Lindsay, 2020).

---

### Official Review · Reviewer_DtEM · 2021-10-29
**Interesting and suitable contribution for this workshop, limited by dataset choice**

**Rating:** 6
**Confidence:** 3

**Review:**

The idea and method of investigation of this study are appealing. It is generally well-written and the methods and analysis are easy to follow. The main limitation, the finetuning on the 10-classes-Imagenette is admitted. For the format of this workshop, and as the results are quite striking Imagenette seems to be a sound and sufficient choice. For extending this work I suggest to consider retraining on ecoset (Mehrer 2021), which is both smaller and has more diverse and ecologically sound classes than ImageNet, and could allow for more specific class-based analyses.

The main issues I find with this work are the following:

* It should be discussed in depth how finetuning an ImageNet-trained network on the 10-classes Imagenette might influence these results. Should there be worry that reducing the number of classes by two orders of magnitude would lead to the late drop in accuracy due to neuronal injury / pruning? The robustness towards node deletion is actually quite surprising.

* A combination of synaptic and neuronal injury should be investigated.

* It is missing details about the finetuning. Have all layers been finetuned or were some left out?

* It is missing details about the weight and node dropout. Likewise, has this been applied equally randomly across all layers?

* While I understand that this work is preliminary, for making the claim that a convnet can model behavioural outcomes of atrophy the results should be connected in more detail to them. E.g., can you pinpoint how you would think the dispersion of the saliency maps is connected to the more specific conditions you mention, like simultanagnosia?

* It should be investigated whether the recognition of a specific subset of classes is more likely to be impaired earlier during the atrophy process, and whether this matches literature about PCA. (Will only be possible after repeating the analysis on a larger dataset.)

---

### Official Review · Reviewer_xSCi · 2021-10-30
**interesting direction, lacking direct comparisons to data**

**Rating:** 6
**Confidence:** 5

**Review:**

The paper investigates the effect of synaptic and neuronal pruning on the ImageNette performance of VGG-19 models and finds that synaptic pruning (setting weights to zero) is much more detrimental to model accuracy than neuronal pruning (setting neurons to zero).

Pros:
* Pushing models towards clinical applications is a novel (and potentially timely) direction
* The paper points to related literature of cognitive deficits due to Alzheimer's disease

Cons:
* Model effects are never directly compared to biological findings. Loose effects of Alzheimer's negatively impacting visual cognition are mentioned, but models are not used to predict / tested on any specific data. Without such direct comparisons, this work loses most of its value in my opinion. Going forward, I strongly encourage the authors to seek out testable data and concretely relate models to data. E.g., are there any estimates of % synapses/neurons damaged and the corresponding deficits to visual cognition? Does local damage to V1 lead to different deficits than damage to V4? The authors mentioned the Brain-Score framework -- such kinds of benchmarks (with synaptic/neuronal damage and corresponding behavioral changes) would greatly strengthen this paper.
* Lack of related work: Cheney et al. 2017 (https://arxiv.org/abs/1703.08245) and Morcos et al. 2018 (https://arxiv.org/abs/1803.06959) found different effects from what I can tell, e.g. the loss in performance was much smoother with synaptic knockout and less sudden with neuronal knockout.

Minor:
* figure 1 is really hard to read, symbols are overlapping so much that they are hard to disentangle. Use e.g. different colors/different symbols/space x-dots further apart.
* I would have liked to see more discussion of how exactly models could be useful to better understand the progression of neurodegenerative diseases.
* To computationally engage with neuroplasticity, we first need to link learning in models to biological learning (e.g. with respect to speed of learning, sample efficiency, etc.) which is ongoing work. Either way, linking to data is also absolutely essential for this research direction.

Overall, I think this project requires some work before making an impact, but I think it could lead to interesting discussions at the workshop that are useful to the audience as well as the authors.

---

### Decision · Program_Chairs · 2021-11-02

Accept (Poster)